# Unsupervised learning reveals interpretable latent representations for translucency perception

**Chenxi Liao** [1] *, **Masataka Sawayama** [2], **Bei Xiao** [3]

**1** Department of Neuroscience, American University, Washington, D.C., District of Columbia, United States of America, **2** Graduate School of Information Science and Technology, The University of Tokyo, Tokyo, Japan, **3** Department of Computer Science, American University, Washington, D.C., District of Columbia, United States of America

* cl6070a@american.edu

**Data Availability Statement:** Human and model data, data analysis code, stimuli, training data, and trained networks are available on Github (https://github.com/cl3789/Translucency-stylegan) and

## Abstract

Humans constantly assess the appearance of materials to plan actions, such as stepping on icy roads without slipping. Visual inference of materials is important but challenging because a given material can appear dramatically different in various scenes. This problem especially stands out for translucent materials, whose appearance strongly depends on lighting, geometry, and viewpoint. Despite this, humans can still distinguish between different materials, and it remains unsolved how to systematically discover visual features pertinent to material inference from natural images. Here, we develop an unsupervised style-based image generation model to identify perceptually relevant dimensions for translucent material appearances from photographs. We find our model, with its layer-wise latent representation, can synthesize images of diverse and realistic materials. Importantly, without supervision, human-understandable scene attributes, including the object's shape, material, and body color, spontaneously emerge in the model's layer-wise latent space in a scale-specific manner. By embedding an image into the learned latent space, we can manipulate specific layers' latent code to modify the appearance of the object in the image. Specifically, we find that manipulation on the early-layers (coarse spatial scale) transforms the object's shape, while manipulation on the later-layers (fine spatial scale) modifies its body color. The middle-layers of the latent space selectively encode translucency features and manipulation of such layers coherently modifies the translucency appearance, without changing the object's shape or body color. Moreover, we find the middle-layers of the latent space can successfully predict human translucency ratings, suggesting that translucent impressions are established in mid-to-low spatial scale features. This layer-wise latent representation allows us to systematically discover perceptually relevant image features for human translucency perception. Together, our findings reveal that learning the scale-specific statistical structure of natural images might be crucial for humans to efficiently represent material properties across contexts.

Figshare https://doi.org/10.6084/m9.figshare.21905463.v1.

**Funding:** The authors received no specific funding for this work.

**Competing interests:** The authors have declared that no competing interests exist.

## Author summary

Translucency is an essential visual phenomenon, facilitating our interactions with the environment. Perception of translucent materials (i.e., materials that transmit light) is challenging to study due to the high perceptual variability of their appearance across different scenes. We present the first image-computable model that can predict human translucency judgments based on unsupervised learning from natural photographs of translucent objects. We train a deep image generation network to synthesize realistic translucent appearances from unlabeled data and learn a layer-wise latent representation that captures the statistical structure of images at multiple spatial scales. By manipulating specific layers of latent representation, we can independently modify certain visual attributes of the generated object, such as its shape, material, and color, without affecting the others. Particularly, we find the middle-layers of the latent space, which represent mid-to-low spatial scale features, can predict human perception. In contrast, the pixel-based embeddings from dimensionality reduction methods (e.g., t-SNE) do not correlate with perception. Our results suggest that scale-specific representation of visual information might be crucial for humans to perceive materials. We provide a systematic framework to discover perceptually relevant image features from natural stimuli for perceptual inference tasks and therefore valuable for understanding both human and computer vision.

## Introduction

Humans assess the appearance of materials every day to recognize objects and plan actions, such as evaluating the ripeness of fruits or stepping on icy roads without slipping. Visually perceiving materials is a first step for anticipating multi-sensory experiences [1–3]. Yet, despite its biological significance and importance in connecting perception to action [4, 5], material perception is poorly understood in both human cognition and artificial intelligence. Materials can be made into objects with any color and shape, and their appearances can be profoundly changed under the joint effect of lighting, viewpoint, and other external factors [6–10], and yet humans can still effortlessly recognize and discriminate materials under diverse contexts [11–13]. How humans infer intrinsic material properties across an enormous range of different contexts remains unsolved.

The challenge of material perception especially stands out for translucent materials such as skin, fruit, wax, and soap. Nearly all materials we encounter permit light into the surface to some degree, which involves a physical process of light transport, namely subsurface scattering (see Supplementary S2 Fig for an illustration) [14, 15]. This gives rise to the essential "translucent" appearance, such as the aliveness of skin. Perceiving translucency not only plays a critical role in material discrimination and identification, such as telling the difference between raw and ready-cooked food, but also allows us to appreciate the beauty of aesthetic objects such as jewelry, sculptures, and still life paintings [16, 17]. Intrinsically, translucency is impacted by the material's optical properties, including absorption and scattering coefficients, phase function, and index of refraction [18–20]. Extrinsically, the object's shape, the surface geometry, and the illumination direction also have striking effects [7, 8, 21–26]. The generative process of translucency involves complex interactions among various intrinsic and extrinsic factors, leading to a wide variety of appearances under different contexts.

There are two main difficulties in studying translucency. First, given the large variation of translucent appearances across materials and scene factors, it has been difficult for humans to provide explicit labels to describe material properties. For instance, the label "soap" can refer

to a variety of translucent appearances and in the meantime, humans may lack precise descriptions for the subtle visual differences between two materials even though they can visually discriminate them. This makes it difficult to measure human translucency perception using real-world stimuli and obtain labeled image datasets based on perception, unlike objects and scenes [27]. The currently available translucent image datasets mostly used rendered images labeled by physical parameters instead of human perception [28]. Second, since many factors affect the appearance of translucent objects, how to systematically discover visual features pertinent to material perception remains unanswered.

For the second difficulty, many previous studies sought to find diagnostic image features for materials using analytical methods. For example, researchers have used well-controlled images to analyze the physics-image relationships of a target material, extract the essential image features, and test if they are diagnostic for human perception [9, 29–37]. Such an approach has been used to study various material qualities, including surface gloss [26, 29, 30, 34, 38–52], surface roughness [53, 54], liquid viscosity [55–58], stiffness of objects [59–62] and cloths [63], surface wetness [64], transparency [65, 66], and translucency [7, 20, 21, 23, 67–72]. However, finding image features from the physics-image analysis can be challenging when a material appears differently across scenes, causing the discovered features to be idiosyncratic to particular scene factors. This problem is especially amplified in translucency (see [9, 37] for reviews). Recently, data-driven approaches have attempted to learn material representations by capturing the statistical structure of material appearance across many image samples [10, 32, 35, 73–78]. These approaches have been used to model human perception. For example, Storrs et al. (2021) rendered opaque gloss and matte images under various illuminations and geometries, trained a variational autoencoder (VAE) model by the images without the supervision of physical attributes, and elucidated the latent image features correlated with human gloss perception [35, 79, 80]. Their work shows the capability of unsupervised learning to disentangle scene factors without physics-image analyses. Some recent works in the perceptual system also utilize this unsupervised approach [77, 81–87]. However, decoding translucency is still challenging because a simple encoder-decoder network used in VAEs cannot disentangle the contributing factors of translucent appearances due to material complexity without the supervision of physical parameters [28].

Here, we aim to learn, unsupervised, a compact latent representation containing the visual characteristics of translucent materials and to explore whether such a latent representation informs perception. We developed a Translucent Appearance Generation (TAG) model trained on our own large-scale dataset of natural photographs of translucent objects, Translucent Image Dataset (TID). We focus on a typical translucent object category commonly seen in daily life, soap. Soaps can be made of different materials and can be manufactured in various shapes and colors, serving as a great medium to investigate the variety of translucent appearances. TAG contains two modules: a style-based generative adversarial network (StyleGAN) [88–90] and a pixel2style2pixel (pSp) encoder [91] (Fig 1A). StyleGAN learns to synthesize images of perceptually convincing translucent materials using its latent space. Unlike the traditional deep generative models (e.g., GAN [92] and DCGAN [93]), StyleGAN utilizes a layer-wise latent space to model high-dimensional distributions of data, leading to an unsupervised separation of visual attributes at different abstraction levels presented in the image domains [88, 94, 95]. More specifically, we use StyleGAN2-ADA, a variant of StyleGAN with adaptive discriminator augmentation (ADA), which inherently applies data augmentations that allow stable training with our relatively small dataset [90]. Meanwhile, we use the pSp encoder to navigate in the learned StyleGAN's latent space and efficiently explore its representative meaning in the expressiveness of translucency (Fig 1B). Our framework provides a pathway to alleviate two difficulties of studying translucency perception. First, without explicit labels, our model learns to represent

**Fig 1. The Translucent Appearance Generation (TAG) model.** (A) Given inputs of natural images, the TAG framework, which is based on the StyleGAN2-ADA generator and pSp encoder architectures, learns to synthesize perceptually convincing images of translucent objects. The model maps photos of translucent objects into the *W+* latent space. The *W+* can disentangle the effects of scene attributes (e.g., shape, material, and body color) and predict the human perception of translucency. (B) The detailed process of embedding a photo into StyleGAN's *W+* latent space. This allows us to generate an image at a particular location in the latent space. (C) Emerged human-understandable scene attributes in the layer-wise latent space. Without supervision, the *W+* spontaneously disentangles three salient scene attributes: material, shape/orientation, and body color. In each row, an original generated image (left) is gradually manipulated by modifying its latent vectors at specific layers. Early-layer ($w_1$ to $w_6$) manipulation of *W+* transforms the shape and orientation of the object. Middle-layer ($w_7$ to $w_9$) manipulation modifies the material appearance. Later-layer ($w_{10}$ to $w_{18}$) manipulation changes the body color (color of the diffuse component of the surface reflection).

materials by finding a candidate distribution of features that is similar to the distribution corresponding to natural images of translucent objects. The learning process is based on a straightforward goal of generating samples that are indistinguishable from the real ones. Second, we obtain a compact representation of material properties from high-dimensional image data. Taking advantage of StyleGAN's representational power, we discover a layer-wise latent space that spontaneously disentangles translucency-relevant attributes and captures the internal dimensions characterizing the variation of translucent appearances, and offer a systematic approach to discover perceptually relevant image features for material perception.

We demonstrate that TAG can create perceptually persuasive and diverse translucent appearances (Fig 1A). Crucially, we show that human-understandable scene attributes emerge in our model's learned latent space (Fig 1C). Without the supervision of physical factors, scale-specific scene attributes related to translucency perception can be separately represented in the layer-wise latent space: material, shape/orientation, and body color. More importantly, we find that the middle-layers of the latent space selectively encode the translucency features and can predict human translucency judgments, while the representations from pixel-based dimensionality reduction methods (e.g., t-SNE, MDS) do not. By leveraging the representational properties of the learned layer-wise latent space, we identify critical image features diagnostic of translucency, such as scale-specific oriented chromatic kernels. Our results suggest

that the unsupervised generative framework may discover an efficient representational space of materials and reveal image regularities potentially used by the visual system to estimate material properties.

## Results

### Unsupervised learning framework: Translucent Appearance Generation (TAG) model

Our main goal is to explore the learned latent space of our model. TAG consists of two parts, illustrated by Fig 1A and 1B: a StyleGAN2-ADA generator [90] and a pSp encoder network [91]. We began by training a StyleGAN2-ADA generator, with unlabeled images from our customized Translucent Image Dataset (TID), which contains 8085 photos of a variety of soaps. StyleGAN2-ADA works with relatively small-sized data, because it utilizes a diverse set of augmentations and an adaptive control scheme to encourage the network to find the correct distribution of data. It demonstrates the ability to capture the statistical structure of images in a variety of image datasets, including artworks of human faces (MetFaces), photos of animal faces (AFHQ CAT, DOG, WILD), and breast cancer histopathology images (BreCaHAD) [90]. Here, TAG's generator network aims to synthesize novel images that are indistinguishable from the real photographs of soaps, without having any knowledge about the physical process of translucency. After training the generator, we could use it to synthesize numerous novel images of translucent objects by sampling from the learned StyleGAN's latent space.

Beyond generating random images of soaps, we are interested in exploring how various visual attributes of material from natural images are represented in the latent space. After obtaining the trained StyleGAN2-ADA generator, we separately trained a pSp encoder network, which could embed a real photograph of soap into the StyleGAN's extended intermediate latent space $W+$. Mapping the real photo into the layer-wise latent space $W+$ leads to accurate reconstruction quality and expressiveness of the input [96–98]. Given a real image, the pSp encoder extracts the 18 latent vectors of $W+$ ($w_1$ to $w_{18}$), which are then inserted into the trained StyleGAN2-ADA generator's convolution layers corresponding to their spatial scales in order to reconstruct the input (Fig 1B). Fig 1A shows examples of the model-generated images of soaps using these methods. The above steps allowed us to effectively examine whether the layer-wise latent space can disentangle the effects of scene attributes on the image appearance and further explore whether such latent representation informs human perception (Fig 1C).

### TAG generates perceptually convincing materials

Before looking into the learned latent space, we first evaluated the perceptual quality of the generated images from two aspects. In Experiment 1, we evaluated the overall image quality and realism of the generated images. In Experiment 2, we further investigated whether the material properties of the generated objects were perceptually convincing and could convey material attributes in the same way as the real images.

In Experiment 1, twenty observers completed a real-versus-generated discrimination task wherein they were instructed to discriminate whether an image is a photograph of soap or was generated from the TAG model. We presented the observers with 300 images of soaps, half of which were real photographs, and the other half were generated images. Fig 2A shows examples of the stimuli. Each stimulus was presented for one second, then the observers made the real-versus-generated judgment (Fig 2B). The 300 stimuli were pre-randomized, and each stimulus was judged twice.

## Experiment 1: Real-vs-generated discrimination

### (A) Examples of stimuli for Experiments 1 and 2

Real photographs

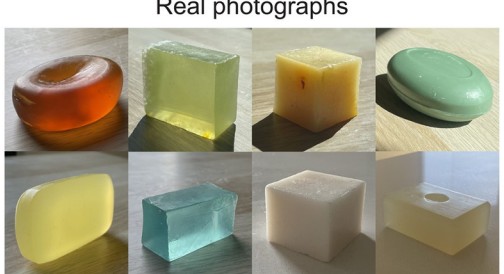

TAG generated images

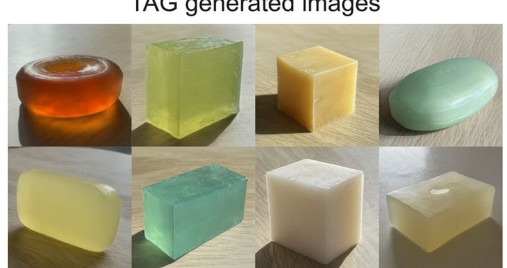

### (B) Experiment 1 interface

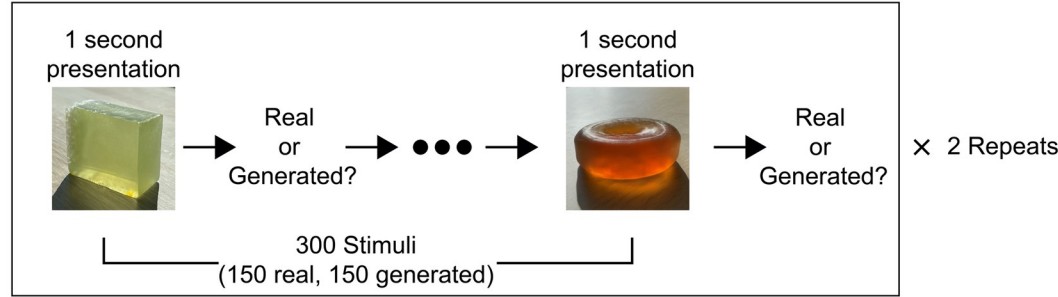

### (C) Observers' judgments on all stimuli (D) Distribution of misjudgment

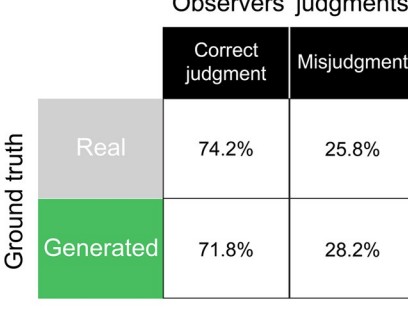

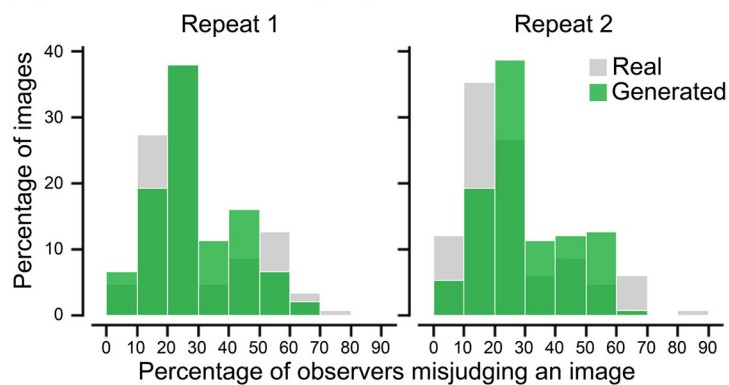

**Fig 2. Experiment 1: Real-versus-generated discrimination.** (A) Examples of real photographs and model-synthesized images of soaps. The "generated" soaps were synthesized by embedding a real photograph into the $W+$ latent space of the trained StyleGAN2-ADA using the pSp encoder. We used 150 real photographs and 150 generated images as stimuli for Experiments 1 and 2. (B) The procedure of Experiment 1. (C) Overall correct and error rates of judging real and generated images. The error rate of 50% indicates pure guessing. (D) Distribution of the percentage of real and generated images misjudged by the observers. The x-axis represents the percentage of observers misjudging an image and the y-axis is the percentage of images being misjudged. Gray color represents data of real images and green represents data of generated images.

If observers could perfectly tell the generated image from the real, they would have 0% misjudgment. On the other hand, if they failed to distinguish between real and generated images, they would be purely guessing and misjudging at a 50% chance. Our results show that across all observers and trials, the observers misjudged 28% of generated images and 25% of real photos (Fig 2C). Meanwhile, Fig 2D shows that the distributions of observers' misjudgments were very similar for both real and generated conditions in both repeats. Specifically, approximately

40% of generated images were erroneously judged as "real" by at least 30% of observers in both repeats (see examples in Supplementary S1 Fig). Only 10% of the generated images were correctly identified by all observers. For a substantial number of images, observers could not discriminate the generated images from the real ones. Our results are on par with the recent findings of human evaluation of StyleGAN-generated high-resolution human face images, where the error rate of judging generated images was also 28% [99]. Overall, the results indicate that our model can successfully generate a large number of perceptually convincing images that fool observers into judging them as real.

In Experiment 2, we evaluated whether the generated images of soaps could convey perceptually persuasive material qualities. Specifically, the same twenty observers from Experiment 1 rated three translucency-related attributes on a seven-point scale (1 means low, 7 means high): translucency, see-throughness, and glow (Fig 3A), which were found in a previous study to be

## Experiment 2: Material attribute rating

### (A) Experiment 2 interface

### (B) Distribution of attribute ratings

### (C) Material attribute ratings for images

### (D) Levels of translucency judged by the observers

**Fig 3. Experiment 2: Material attribute rating.** (A) The user interface of Experiment 2. (B) The distribution of the mean normalized attribute ratings across observers. For each observer, we normalize their attribute ratings to the range of 0 to 1. The x-axis represents the normalized ratings of an attribute averaged over 20 observers, and the y-axis shows the percentage of images. (C) The scatter plots of ratings between a pair of material attributes, with the Pearson correlations shown at the top. All correlation coefficients are statistically significant at the confidence level of 95% ($p < 0.001$). In both (B) and (C), gray and green colors represent results for real and generated images, respectively. (D) Examples of real and generated images judged to have different levels of translucency. We grouped the images based on the mean normalized translucency rating: high (0.6 to 1), intermediate (0.2 to 0.6), and low (0 to 0.2).

descriptive in semantic judgments of translucent objects [12]. Material attribute ratings were normalized to the range of 0 to 1 for each observer. For each image, the normalized attribute ratings were averaged across observers, and subsequent data analysis was based on these values. Fig 3B shows that observers perceived different degrees of translucency, see-throughness, and glow from the generated images, with ratings distributed similarly to those of real photos. This shows that observers could perceive a wide range of translucent material attributes from the generated images. Meanwhile, the material attributes perceived by the observers are highly positively correlated with one another for both real photographs and generated images of soaps (Fig 3C). The correlations among the attributes are in agreement with our previous empirical findings [12]. Fig 3D shows examples of real and generated images judged to have various degrees of translucency similar to that of real photographs. Together, our results suggest that TAG learns to synthesize diverse perceptually convincing translucent appearances and conveys material attributes similar to real photographs.

Meanwhile, we investigated the plausibility of learning to generate diverse translucent appearances using an alternative GAN framework, the Deep Generative Adversarial Network (DCGAN). Unlike the multiple-scale latent representation used in StyleGAN, DCGAN uses a single input latent space and fractionally-strided convolutions in its generative process. Although DCGAN captured some rough visual impressions implying translucency at the relatively coarse image resolution, it is limited in synthesizing more nuanced details for perceptually convincing translucent qualities. However, the observation that the DCGAN's generative results at 64 pixels × 64 pixels resolution could already show some translucent characteristics provides empirical evidence that translucency impressions could be conveyed in a more compressed form (see Methods and Supplementary S8 Fig).

## Emergence of perceptually meaningful scene attributes in the learned latent space

What makes the generated images convey perceptually persuasive material appearance? We hypothesize that TAG's $W+$ latent space is obliged to learn the explanatory factors underlying the structure of observed data. To test our hypothesis, we systematically manipulated different layers of the latent code and inspected how these manipulations affect the visual attributes of the output image. Specifically, we applied morphing between the latent codes of a pair of images (a source and a target), which differ in their shapes, intrinsic materials, lighting environments, and body colors (Fig 4A).

Given two generated images A (source) and B (target) with their corresponding $W+$ latent codes, $w_A$ and $w_B$ ($18 \times 512$-dimensional latent codes), which could have arbitrary visual characteristics, morphing can be applied on particular layers of their latent codes to create a sequence of generated images with visual appearances lying between the source and the target. The morphed latent vectors are generated by a linear interpolation of a particular set of layers ($s$) between the source ($w_A^{\{s\}}$) and target ($w_B^{\{s\}}$) while keeping the other layers from source image unchanged:

$$w_\lambda^{\{s\}} = (1 - \lambda)(w_A^{\{s\}}) + \lambda(w_B^{\{s\}}), \lambda \in [0, 1] \tag{1}$$

where $\lambda$ is the interpolation step and $w_\lambda^{\{s\}}$ is the resultant latent vectors of the set of layers. The generator then uses the combination of $w_\lambda^{\{s\}}$ and the remaining unchanged latent vectors from the source image to produce a new image (e.g., one of the intermediate images in the image sequence shown in Fig 4A). For example, when we apply morphing between the source and the target on their latent vectors of layers 7, 8, and 9, the resultant latent vectors follow

# Experiment 3: Perceptual evaluation of emerged scene attributes in W+

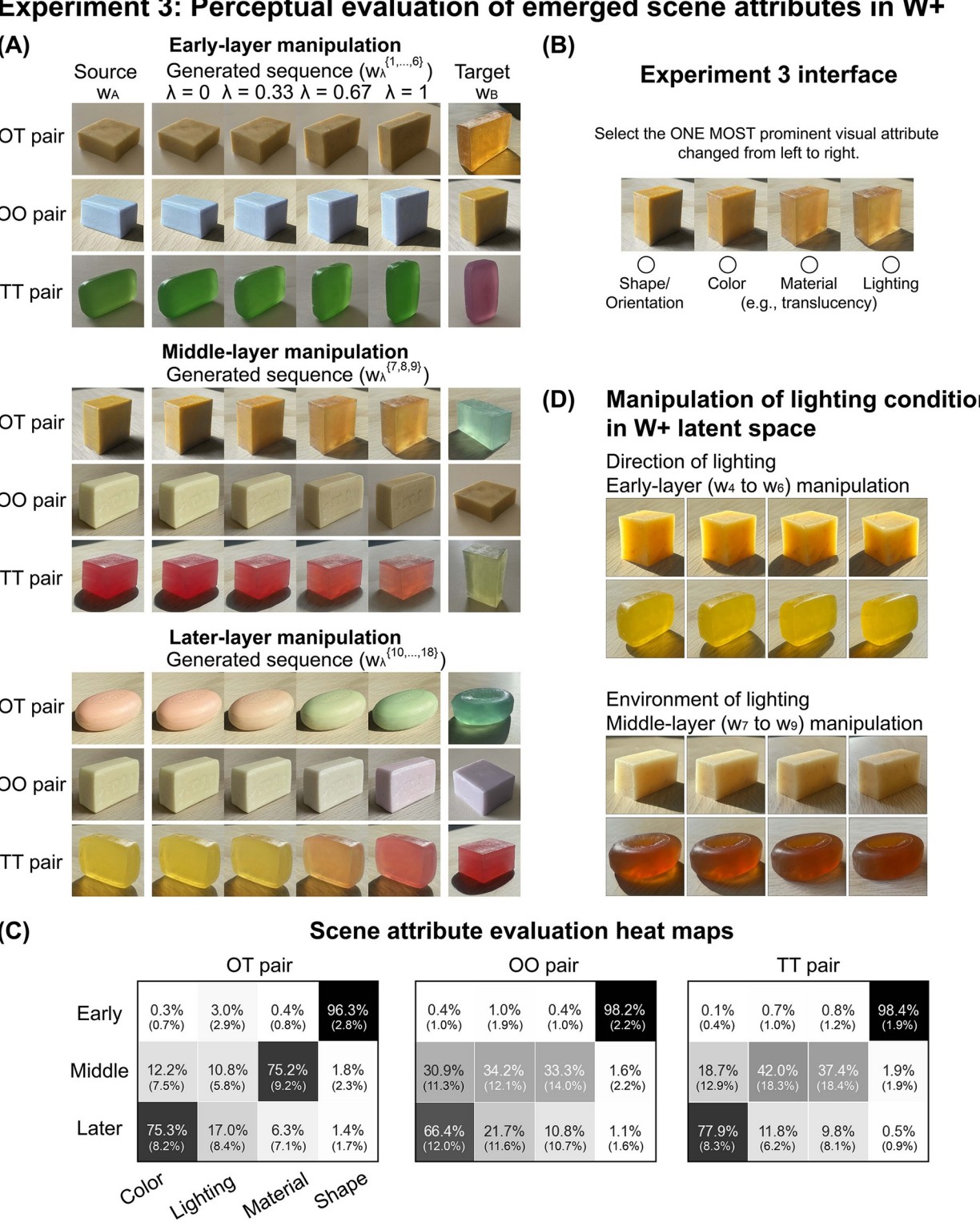

**Fig 4. Experiment 3: Perceptual evaluation of emerged scene attributes in the layer-wise latent space.** (A) Examples of morphed image sequences used in Experiment 3, generated by linearly interpolating between the latent codes of source ($w_A$) and target ($w_B$) separately at the early ($w_1$ to $w_6$), middle ($w_7$ to $w_9$), and later-layers ($w_{10}$ to $w_{18}$). The λ is the interpolation step from the source image in the linear interpolation. Source-target pairs were picked under three conditions based on soap's material properties: opaque-translucent (OT), opaque-opaque (OO), and translucent-translucent (TT). (B) The user interface of Experiment 3. (C) The perceptual results on how different layers correspond to scene

attributes. The number in each cell represents the average percentage of times observers chose a visual attribute as the most prominent one that changes in the image sequence generated by the corresponding layer manipulation. The standard deviation across observers is shown in parentheses. Each row of the heat map accounts for 50 image sequences. (D) The representation of lighting in the latent space. Top panel: manipulation of early-layers (layers 4 to 6) also changes the direction of lighting. From left to right, the lighting direction rotates clockwise. Bottom panel: manipulation of middle-layers (layers 7 to 9) alters the environment of lighting. From left to right, the strength of backlighting gradually decreases.

$w_\lambda^{\{7,8,9\}} = (1 - \lambda)(w_A^{\{7,8,9\}}) + \lambda(w_B^{\{7,8,9\}})$. When $\lambda = 0$, the output is the original latent vectors of the source image. When $\lambda = 1$, the latent vectors on layers 7, 8, and 9 of the source image are replaced by those of the target image (Fig 4A middle panel).

Fig 1C shows examples of the layer-wise manipulation in the $W+$ latent space. We observed the emergence of three salient attributes when we performed image morphing at different layers: early-layers (layers 1 to 6) determined the shape and orientation of the soap, middle-layers (layers 7 to 9) effectively changed the material (e.g., transformed from glycerin to milky soap, and vice versa), and later-layers (layers 10 to 18) primarily changed the body color of the object. This shows that StyleGAN's deep generative representation mechanistically disentangles the scene attributes without external supervision.

## Perceptual evaluation of emerged scene attributes

To examine how naive observers interpret the scene attributes that emerged in the layer-wise representation of $W+$ space, we created image sequences by morphing between selected pairs of images on three different sets of layers (early, middle, later). For each layer manipulation (Eq 1), we selected source-target image pairs under three material conditions: opaque-translucent (OT), opaque-opaque (OO), and translucent-translucent (TT). Together, we sampled 450 image sequences (see Methods). Fig 4A shows examples of image sequences generated from the three layer-manipulation methods (top panel for early-layer manipulation, middle panel for middle-layer manipulation, bottom panel for later-layer manipulation) under three source-target material conditions. For each image sequence, observers were asked to select the "ONE MOST prominent visual attribute changed from left to right" (Fig 4B). Fig 4C illustrates which attribute observers chose as the dominant change for each layer manipulation. The heat maps show that observers unanimously agreed that manipulation on early-layers changed the shape of the objects (approximately 97%), independently of the material condition of the target and source images. Observers also agreed that manipulation on middle-layers mainly changed the translucent appearance of the objects (approximately 75%) for opaque-translucent pairs. When the source and the target have similar materials (OO and TT pairs), the middle-layer manipulation led to a less obvious change of material appearance (approximately 35%), and observers also selected lighting or color as the main variation factor depending on the scene. For example, when we morphed two translucent soaps, either material or lighting could be viewed by observers as the dominant change (Fig 4A, middle panel, third row). Lastly, observers mostly agreed that manipulation on later-layers changed the body color of the objects across material conditions (approximately 73%). We conducted a Bayesian multilevel multinomial logistic regression on the behavioral data, and analysis results coincided with our observations [100]. All three layer-manipulation methods are credible parameters for the estimation of the most prominent scene attribute. We also examined the conditional effects of layer manipulations. For the early-layer manipulation, the estimated probability of selecting "shape/orientation" was close to 1 across all three types of source-target pairs. For the middle-layer manipulation applied on OT pairs, the estimated probability of selecting "material" was 77.9% (95% highest density interval, [69.5%, 84.5%]) (Supplementary S3 Fig). These results show that

the scene attributes disentangled in the latent space are perceptually meaningful and each attribute can be separately controlled in different layers' latent vectors.

We also observed some participants chose lighting as the dominant change resulting from the middle-layer manipulation for similar target and source materials, suggesting that the middle-layers of the latent space can also represent lighting to some degree. The effect of lighting may have two aspects: the direction and the environment of lighting. The direction of lighting, expressed in the images through the position and shape of the cast shadow, was captured in a subset of earlier layers (layers 4, 5, and 6). The top panel in Fig 4D shows manipulating such layers conveyed the impression of rotating the light source clockwise. On the other hand, the environment of lighting (e.g., sunny versus overcast) affects the color distribution of objects in an image. This effect is manifested in the middle-layers (layers 7 to 9). The bottom panel in Fig 4D shows that manipulating these layers yielded the impression of varying the strength of backlighting. This observation is consistent with previous findings that the lighting environment affects translucent material perception in that objects under strong backlighting tend to appear more translucent [7, 21].

## The middle-layers of the latent space capture human translucency perception

Our next goal was to examine whether the middle-layers of the latent space could capture human translucency perception. To derive quantitative translucency prediction from the model, we trained a linear support vector machine (SVM) classifier to find the decision boundary with each layer of the images' latent codes that best distinguishes translucent soaps from opaque ones. We manually labeled the soaps into two categories based on their listed ingredients: milky and glycerin. We sampled 1000 real photos from the TID dataset. Half were glycerin soaps, and the other half were milky soaps. The shape, lighting condition, and body color varied significantly across instances. After obtaining the corresponding $W+$ latent codes of the embedding of real photos through the pSp encoder, we extracted their latent vector at each of the 18 layers to train a linear SVM classifier. Therefore, we had 18 distinct decision boundaries. Fig 5A illustrates the trained decision boundary ($d_i$) using the $i$-th layer of $W+$.

Next, we computed the SVM model predictions and compared them with the human attribute ratings measured in Experiment 2. Specifically, we obtained 18 distinct model prediction values from each layer's latent vector for the 150 generated images used in Experiment 2. For a given image with its $i$-th layer latent vector, we measured its distance from the learned decision boundary $d_i$ (normalized to 0 to 1 range). For example, as shown in the middle columns in Fig 5B, using an image's layer-9 latent vector, we could plot its model prediction value against the mean normalized human attribute rating for translucency, see-throughness, and glow, respectively. The Pearson correlation between the model prediction and perceptual rating ($r_{hc}$) is calculated for each attribute. The data shows that predictions from a middle layer (e.g., layer-9) strongly correlate with human material attribute ratings while the predictions from an earlier layer (e.g., layer-6) and a later layer (e.g., layer-12) have a relatively weak correlation with perception. By repeating this step for each layer, we obtained the correlation coefficients between each layer's model prediction and the perceptual ratings (see details in Supplementary S2 Table). Fig 5C shows the tuning curve of correlation coefficients over the layers. The correlations $r_{hc}$ peaked at the middle-layers (layers 7, 8, and 9), implying that these layers may most effectively encode visual information that observers utilized for translucency perception.

The trained SVM serves as a general guidance for material appearance editing. The decision boundaries from middle-layers reflect the linear separability of intrinsic material in the latent space. The normal to the decision boundary becomes an interpretable direction that captures

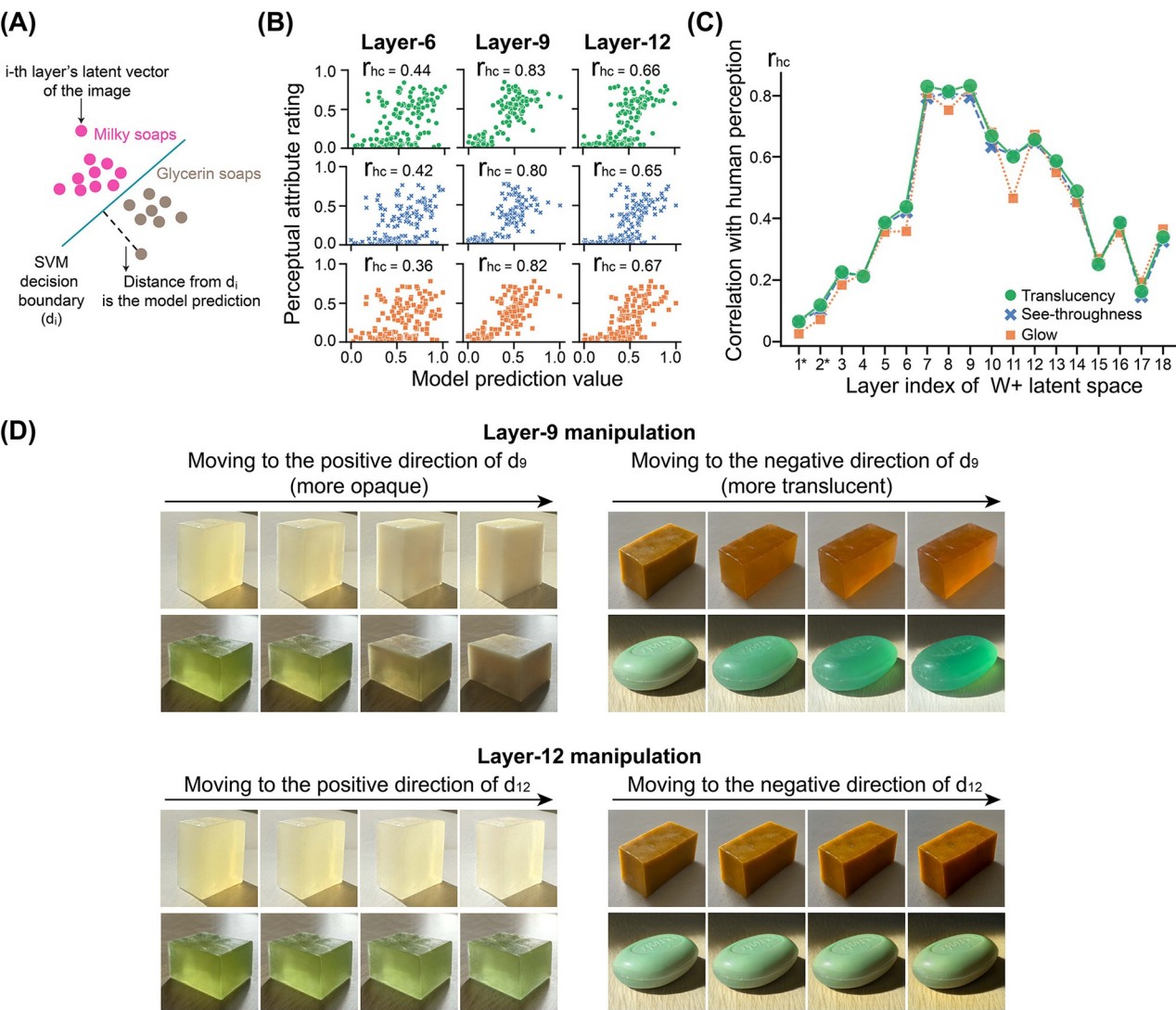

**Fig 5. The middle-layers of *W+* latent space can effectively modulate translucency of generated images and predict human perception.** (A) Illustration of a trained layer-specific supported vector machine (SVM) classifier for the milky-versus-glycerin soap discrimination. (B) The scatter plots show the model prediction values versus the human mean normalized attribute ratings for each generated image in Experiment 2. Green, blue, and orange colors represent the data for translucency, see-throughness, and glow, respectively. (C) The tuning curve of correlation coefficients (correlation between model prediction and human perceptual rating, $r_{hc}$) over all layers in the *W+* latent space. Model prediction values using the middle-layers' decision boundaries ($d_7$, $d_8$, and $d_9$) strongly correlate with human attribute ratings. "*" indicates the correlations at that layer are statistically insignificant at the 95% confidence level. (D) Examples of translucency-modulated sequences. Top: Manipulating the layer-9 latent vector of the original image (left end) along the normal of the learned decision boundary has a coherent effect on the translucent material appearance of the object. Left: Moving to the positive direction of the normal of the decision boundary makes the soap appear more opaque. Right: Moving to the negative direction of the normal of the decision boundary makes the soap appear more translucent. Bottom: Manipulating the layer-12 latent vector of the original image along the normal of the learned decision boundary does not fundamentally change the translucent appearance.

the variation of material appearance. As shown in Fig 5D's top row, manipulating the layer-9's latent vector of a given image (left end) along the positive direction of the normal to $d_9$ persuasively made the material more milky and opaque, without changing the object's shape. Conversely, moving to the negative direction made an opaque soap more translucent. In contrast, manipulating a single latent vector from early or later layers (e.g., layer-12) along the found decision boundary's normal did not lead to effective modification of the material appearance

(Fig 5D bottom). The manipulation on all layers can be found in the Supplementary S4 and S5 Figs.

As controls, we computed the embeddings from the raw images (i.e., pixel values from the R, G, B channels) of translucent objects by applying dimensionality reduction methods: t-distributed stochastic neighbor embedding (t-SNE) and multidimensional scaling (MDS). Instead of using the latent code of the image in $W+$ space, we used the embedded spaces obtained from raw images to train a SVM model for milky-versus-glycerin classification. For each method, we created a 512-dimensional embedding of the same 1000 real photos of "milky" or "glycerin" soaps used to train the 18 layer-wise SVM classifiers, as well as the 150 TAG-generated images used in Experiment 2. Next, upon training the SVM classifier on the corresponding pixel-based embedding of 1000 photos, we computed the normalized distance from the trained decision boundaries as the model prediction value for each of the 150 TAG-generated images, and the correlations $r_{hc}$ with human perceptual ratings. In contrast to the middle-layers (Fig 5B), the predictions from neither t-SNE nor MDS significantly correlate with human psychophysics ($p > 0.1$) (see S9 Fig in Supplementary).

## Translucency features are established at the mid-to-low spatial scales

To break down how translucent appearance is created in the final output, we examined feature maps generated in the intermediate stages of the synthesis network of StyleGAN2-ADA [88, 101]. The generator starts from a learned constant input of size $4 \times 4 \times 512$ and gradually expands the spatial resolution via affine transformation layers. At each resolution, from $8 \times 8$ to $1024 \times 1024$, an additional single convolution layer (tRGB layer) transforms the feature maps into the RGB image. As shown in Fig 6, we visualized the intermediate steps to generate the images of soaps with their corresponding $W+$ latent codes.

Consistent with the discovery of emerged scene attributes in the latent space, the early-layers ($w_1$ to $w_6$), including $8 \times 8$ to $16 \times 16$ resolutions, formed the general shape and contour of the object. The middle-layers, with layers 7 and 8 ($w_7$ and $w_8$) at $32 \times 32$ resolution and layer 9 ($w_9$) at $64 \times 64$ resolution, established the critical features of translucency. The image contrast and color variation across the volume of the soap in the $64 \times 64$ resolution images gave the impression of "glow", which is useful to distinguish translucent materials from opaque ones. At $128 \times 128$ resolution (layers 11 and 12), surface reflectance properties such as specular highlights and caustics were further specified. The later layers ($w_{13}$ to $w_{18}$), from $256 \times 256$ to $1024 \times 1024$ resolutions, enriched the details of the lighting environment and color scheme, delivering more appealing material appearance. This suggests that latent image features at relatively coarse spatial scales are sufficient to capture the visual impression of translucent materials.

## Diagnostic image features for translucency

To understand what information the intermediate generative representation encodes, we explored the image descriptors for the tRGB layer's representation with the middle spatial scale, which is sensitive to translucency (Fig 7). Inspired by sparse coding used in understanding natural images [102–105], we applied independent component analysis (ICA) [106] on local regions of the intermediate tRGB images to investigate the efficient representation of translucent appearances. Specifically, based on the results of Experiment 2, we created a new set of high-translucency generated images and extracted the intermediate tRGB images with 64 pixels × 64 pixels, whose layer is sensitive to translucency emergence (see Methods). While keeping the relative kernel size constant with the StyleGAN's convolution process, we applied FastICA to learn 64 basis functions [106].

## Image synthesis process visualized by intermediate tRGB layers

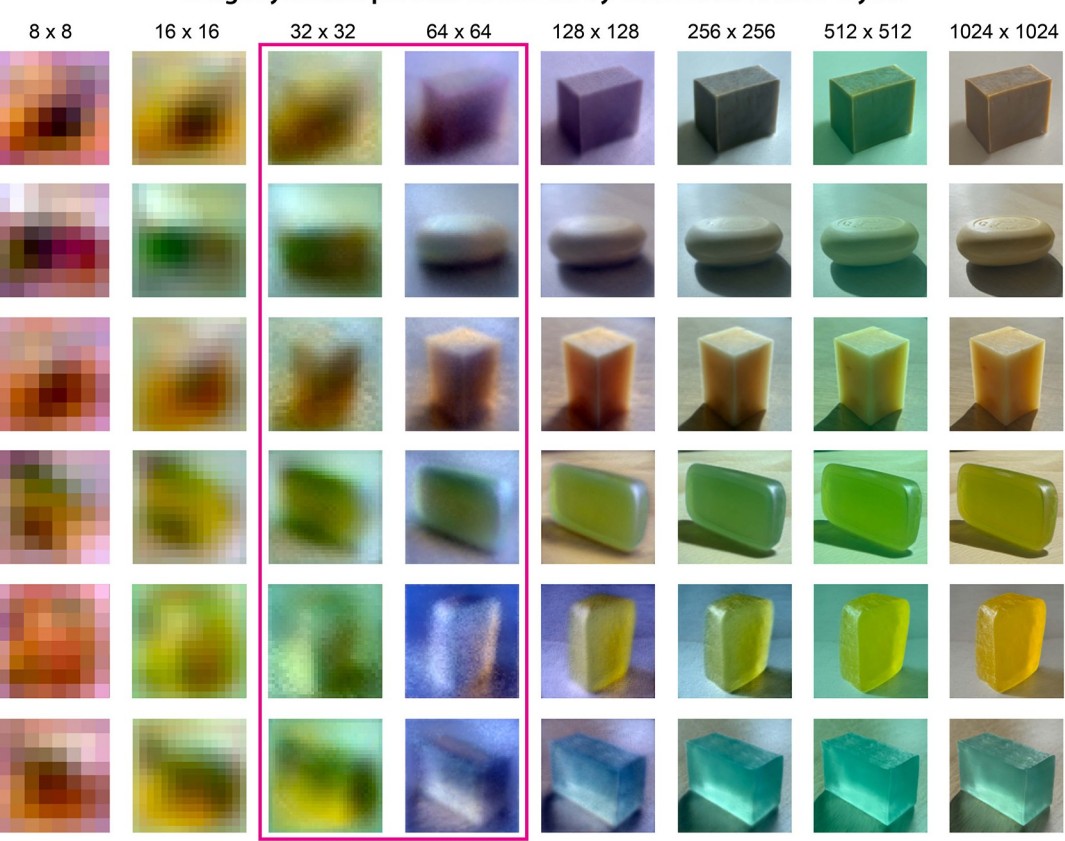

**Fig 6. Visualization of the generative process of the network.** Impression of translucency emerges at the early stages of the image synthesis process while more details of the appearance are added in the later stages. Each row corresponds to the intermediate generative outputs from a sequence of tRGB layers at different spatial resolutions in StyleGAN2-ADA's generative network. Translucency-related features are established as early as 32 pixels × 32 pixels (layers 7 and 8) and 64 pixels × 64 pixels (layer 9). The surface reflective properties such as specular highlights are only clearly visible at 128 pixels × 128 pixels (layers 11 to 12). The body color of the soap was finalized at the resolution of 1024 pixels × 1024 pixels (layers 17 to 18).

In the learned representation (Fig 7B), the activation features are chromatic or achromatic with a variety of orientations. Fig 7C demonstrates the results of applying the three-dimensional convolution of each of the 64 kernels to a real photograph of translucent soap. While luminance kernels provide information of object contours, shadow boundaries, and specular reflectance, chromatic kernels reveal subtle image features indicating translucency, such as color gradients around the edges and corners. Fig 7D shows examples of filtering results on a few translucent and opaque objects. For example, applying the oriented chromatic kernels (rows 1–4 in the matrix of convolution results) on the transparent soaps (columns 1 and 3) activated patterns of color variations on the caustics, which are not present in the more opaque soap (rows 1 and 2, column 2). Next, the red-green chromatic kernels also detected the internal "glow" of the translucent soaps. For example, the convolution results on the yellow milky translucent soap (column 4) showed the spatial gradient of saturation near the edges (row 1, column 4). At the same time, the resulting image also revealed the "glowing edge" on the same soap (row 1, column 4). Notably, the orientation-free chromatic kernels revealed the color variation over a relatively coarse spatial scale across the object, which might be diagnostic of translucency (row 5, column 4). Furthermore, these translucency-related features could not be

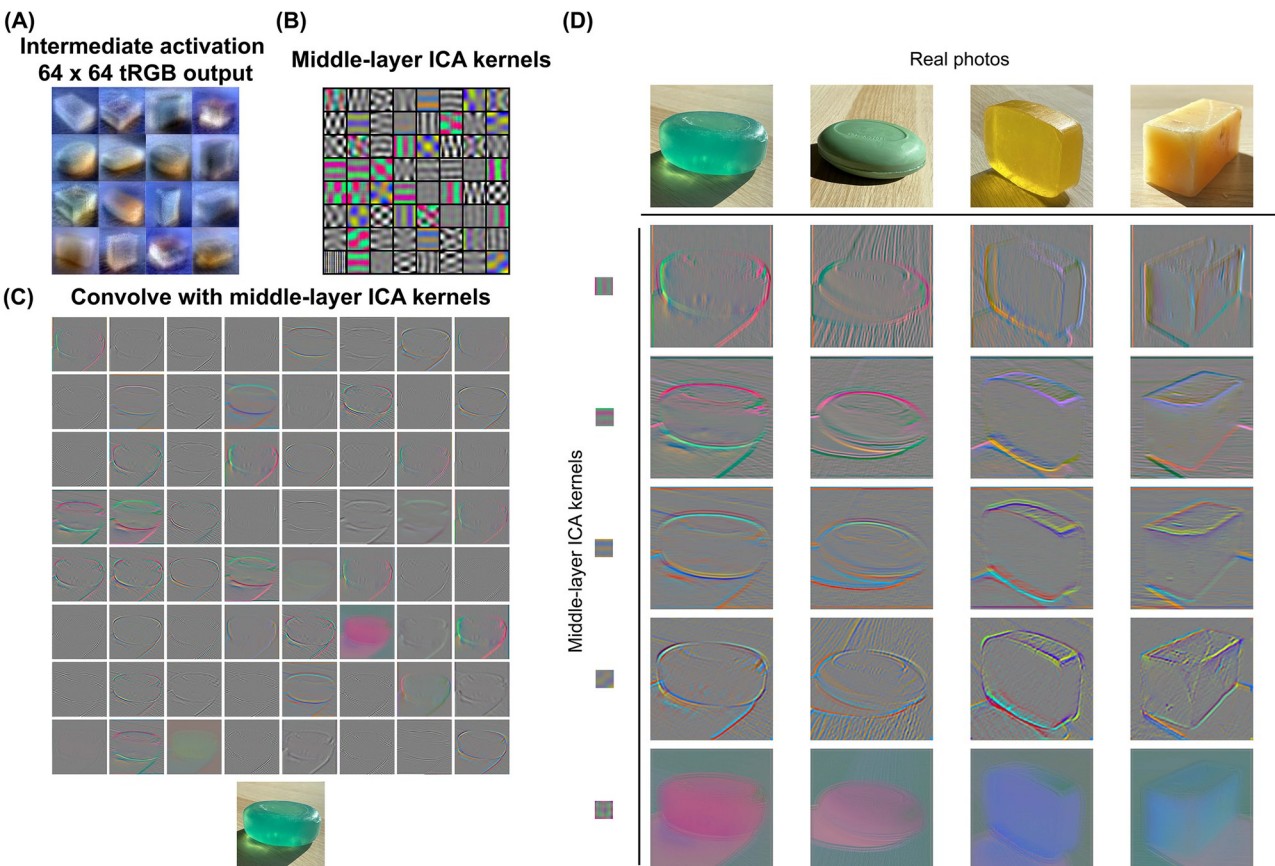

**Fig 7. Visualization of features for translucency.** (A) The intermediate generative results (tRGB layer output at 64 pixels × 64 pixels resolution) of the images from the high-translucency dataset. The images are resized for display. (B) Middle-layer ICA kernels obtained by training a system of 64 basis functions on 24 pixels × 24 pixels image patches extracted from images in (A). The kernels are of size 24 × 24. (C) Visualization of applying three-dimensional convolution of the individual ICA kernels in (B) on a real photograph of translucent soap. (D) The resulting filtered images of four different soaps with selected chromatic kernels. The mid-to-low spatial frequency chromatic kernels can capture features of translucency such as "chromatic caustics" (row 2, column 1), "glowing edges" (row 1, column 4), and "inner glow" (row 1, column 1 and 4). The orientation-free kernel in the last row reveals the variation of color over a relatively coarse spatial scale, which is also diagnostic of translucency.

obtained by the basis functions extracted from the coarser intermediate representation (Supplementary S7 Fig). Together, our results indicate that the oriented chromatic kernels with mid-to-low spatial frequency can be diagnostic for translucent appearance.

## Discussion

We presented a deep image generation model trained with natural photographs to obtain a compact layer-wise latent space that can capture the human perception of translucency. Our study demonstrates that the learned latent space spontaneously disentangles salient visual attributes and captures the latent dimensions of translucent appearances. Notably, we find the represented scene attributes are scale-specific, where early-layers represent shape, middle-layers represent translucency, and later-layers represent body color. The middle-layers of the latent space can successfully predict the human perception of translucency in various generated images. Our findings suggest that humans might use a scale-specific structure to characterize visual information from retinal images, facilitating the representation of materials for estimating their attributes under various contexts. Our framework could serve as an effective method

for discovering generalized image features across a high degree of perceptual variability of materials.

The image generation process of our model (Fig 6) resembles the strategy an artist uses to paint a translucent object by structurally depicting the observed visual attributes. Therefore, the representational system the model learns might be similar to those of the mental process of painting. The Dutch artists in the 17th century were capable of painting vivid translucent objects on the canvas by depicting the critical image features that trigger a visual impression of translucent properties, without strictly conforming to physical laws [17, 107]. Imagine an artist painting a grape on a dining table (e.g., *Still Life with Oysters and Grapes, Jan Davidsz. de Heem, 1653*). As a first step, before exquisitely adding any details, the artist often starts with carving out the object's contour. After the general shape is set, colors are gradually filled in to mimic shadow and shading to yield a first impression of the 3D shape and reflect the lighting condition in the scene. More details are added to depict surface reflections and caustics. The artist can continually perfect the painting by adding fine details to deliver a more convincing material quality. The combination of multiple levels of detail in the painting contributes to the formation of translucency appearance. Our TAG model generates realistic translucent appearances in a similar scale-specific manner. In contrast, a generative model lacking the multi-scale representational power (e.g., DCGAN) may not exquisitely represent the complex visual phenomena of translucency, despite it learns some rough visual impressions. Given that previous studies have also shown that the scale-specific process has a role in material perception [108–110], discovering and formulating the structure of visual information in a scale-specific manner might be helpful for recoding the complexity of material appearances to obtain an efficient representation.

In our study, we use soap as a medium to illustrate the possibility of learning a semantically meaningful representation of natural images of materials. Although it is possible that the exact meaning represented by the latent space differs across the training datasets, the corresponding latent space will still disentangle scene attributes with various abstract levels arising from the scale-specific image features. We expect the middle-layers (medium coarseness) to represent volumetric material appearance even if the model is trained with image datasets of other materials. For our dataset, we demonstrated that scale-specific image features can be separately controlled. The translucent appearance (i.e., associated with middle resolution features) and the body color (i.e., associated with fine resolution features) can be directly manipulated without changing the shape of the object (i.e., associated with coarse resolution features). The observations from our finding also align with previous investigations of StyleGAN's representative power of its layer-wise latent space. In the generation of images of human faces, "styles" of coarse spatial resolutions (2 to 4 cycles/image) correspond to high-level aspects such as pose and face shape, "styles" of middle resolutions (8 to 16 cycles/image) control smaller scales of facial features and hairstyle, and "styles" of fine resolutions (32 to 512 cycles/image) contribute to microstructures and color scheme [88]. Likewise, in indoor scene synthesis, the latent space could separately control the spatial layout of the room (coarse), categorical objects in the scene (middle), and color scheme (fine) [94]. Constrained by the diversity in geometry and material appearance of the current training samples, our TAG model might not be directly used to synthesize images of another material that has more irregular geometries, such as crystals. However, learning about the soaps is a starting point to capture essential translucent appearances in the real world. Therefore, in future works, it is possible to extend our model to learn the representation of image data across broad categories of materials with techniques such as transfer learning.

Our study constitutes a break from the long history of studying material perception using well-controlled computer-rendered images. We discovered critical image features that are

diagnostic of translucency across diverse geometries and lightings by applying unsupervised learning schemes on a large-scale dataset of natural photographs of translucent objects without specifically constrained physical environments. Some of our found image features could be robust indicators for translucent materials, and they confirm previous empirical findings. For instance, the edge intensity profile on translucent objects has been found to be different from those of opaque ones [111]. Our ICA analysis shows that oriented chromatic kernels can detect complex patterns along the translucent edges (e.g., Fig 7D row 2, column 4). In addition, such chromatic kernels also capture the effect of "glow", an important feature characterizing the spatial distribution of color of translucent materials [12, 21]. Furthermore, our results indicate that the presence of caustic patterns can be an important cue for translucency perception [112].

We also discover the intricate role of color in translucent appearances. Most of the previous works explored the effect of color on material perception and recognition by manipulating the color/luminance distribution of material images [12, 21, 64, 71, 113, 114]. For example, converting color translucent images to grayscale ones decreases perceived translucency [12, 21, 64, 72]. However, it is still unclear how the visual system functionally processes color information for material perception. Our findings, based on a data-driven approach, suggest that two functional aspects lie in color translucency processing: body color and spatial color processes. The body color represents the color of the matte component of surface reflectance, which is usually determined by the color of the dye used to make the soap. The latent space in our model can represent the body color of soap separately from the material appearance. By manipulating the middle-layers of the latent code, we can create images of objects with different types of translucent appearances but of similar body color. This suggests the model can establish a translucency impression without varying the body color. The other aspect is the spatial variation of color over the volume and surface of an object (e.g., the color gradient within an object due to light scattering and absorption). This "spatial color" is crucial for providing the translucent appearance in the middle-layers (Fig 1C, top panel) and can be detected by the chromatic kernels with the mid-to-low scale (Fig 7D). Notably, this color process is scale-specific, i.e., a coarser kernel cannot detect the spatial color variation of translucency (Supplementary S7 Fig). Furthermore, the spatial color can be independent of the white-balancing process because the middle-layers in our model do not fix the white point in the scene (Fig 6). The finding suggests that the processing of saturation and hue based on a white point may not be necessary for this spatial color process. As such a spatial color process has been little understood in color vision literature [115, 116], our work might provide novel directions for probing the role of color in material perception and other high-level visual processing in the brain.

The deep generative network (StyleGAN) is not designed to emulate biological vision systems, even though the elementary functional mechanisms (e.g., convolution, nonlinearity) are inspired by biological brains [117–120]. Therefore, we do not assume the learning process of StyleGAN is necessarily the mechanism of human material perception. Here, we take StyleGAN's representative power to model the feature space of diverse material appearances, compare it with psychophysics, and discover the latent image features that humans might have used to estimate material properties in natural scenes. We also acknowledge that the image features we learned in our model are still considered early-to-mid-level visual information. Future models need to be developed to address the role of top-down influence, such as context, object identity, and individual experience, on material perception. Nevertheless, a generative model, such as StyleGAN, could be conceived as an apparatus that effectively simulates images of morphologically controllable materials, serving as an additional data source that supplies a compressed copy of the real image data of materials [121].

One extension of the current work is to use our stimuli to measure brain responses to translucent material properties. One large obstacle to probing the neural correlates of material perception has been the lack of an effective way to manipulate the stimuli that isolate the effects of various external factors on material appearance while keeping the image's appearance natural and realistic. Our material manipulation through the latent space illustrates a novel and efficient approach for conditionally creating stimuli with translucent appearances resulting from a specific combination of scene attributes. Moreover, the discovered latent representation can be valuable for encoding/decoding investigations in brain-imaging studies to probe the interaction between neural representations of 3D shape, color, and materials, thus providing an efficient tool to discover the neural correlates of material perception. More generally, the approach we take here—using StyleGAN to derive a latent representation for translucency perception—is widely applicable to discover perceptually relevant features for a variety of visual inference tasks that deal with complex physical stimuli.

## Methods

### Ethics statement

All psychophysical experiments were conducted in accordance with the Declaration of Helsinki, with prior approval from the American University. All the experimental designs involving human participants were approved by the Institutional Review Board at American University.

### Translucent Image Dataset (TID)

Our customized image dataset of translucent objects has 8085 photographs of soaps. The dataset was created by photographing a variety of real-world soaps in natural backgrounds. We collected 60 unique soaps that included diverse materials, geometries, surface relief, and colors. We used an iPhone v12 mini smartphone to photograph our collection of soaps under various lighting environments and viewpoints at a relatively fixed distance, and built a dataset of high-resolution images (1024 pixels × 1024 pixels JPEG images). In each photograph, the object was centered in the image. We did not intentionally/precisely balance the dataset on the distribution of shape, body color, lighting environment, and viewpoint. Approximately, our dataset covers a variety of illumination directions: backlighting (about 44%), partial-front lighting (about 8%), side lighting (about 40%), and diffuse (dim) lighting (about 8%). To our knowledge, this is the first large-scale natural image dataset of translucent materials and one of few image datasets of real-world materials.

### Unsupervised learning framework: Translucent Appearance Generation (TAG) model

**Deep generative network StyleGAN2-ADA.** We trained StyleGAN2-ADA, on the TID dataset using the TensorFlow implementation of the model available at https://github.com/NVlabs/stylegan2-ada. StyleGAN2-ADA consists of two networks trained through a competitive process: a style-based generator, and a discriminator. The generator creates "fake" images, with the aim of synthesizing realistic images of soaps. The discriminator receives both "fake" and real images, and aims to distinguish them. As the training progresses, both the generator and the discriminator improve until the "fake" images are indistinguishable from the real ones. The training of the style-based generator involves two latent spaces. There is an input latent space $Z$ that is normally distributed. Hence, a sequence of eight fully-connect layers transforms $Z$ to an intermediate latent space $W$. The dimensions for both $Z$ and $W$ spaces are

512. With the 1024 pixels × 1024 pixels output, the generator starts with a constant input of size $4 \times 4 \times 512$ and gradually adjusts the "style" of the image at each of 18 convolution layers based on the latent vector [88]. For every major resolution (every resolution from 4 pixels × 4 pixels to 1024 pixels × 1024 pixels), there are two convolution layers for feature map synthesis and a single convolution layer (i.e., tRGB layer) that converts the output to an RGB image. Weight modulation and demodulation are applied in all convolution layers, except for the output tRGB layers [89]. At each convolution layer $i$, the generator receives the input through "style", which is a learned affine transformation from the 512-dimensional latent vector $w \in W$. More explicitly, when generating an image from $W$ space, the same vector $w$ is used for all convolution layers.

Using the network architecture of StyleGAN2, StyleGAN2-ADA inherently applies a wide range of augmentations on the input data to prevent the discriminator from overfitting, while ensuring that none of the augmentations leak to the generated images. During training, each image is processed by a series of transformations in a fixed order, and each transformation is randomly applied with probability $p \in [0, 1]$, which is adaptively adjusted to counter the effect of overfitting. This variant is named Adaptive Discriminator Augmentation (ADA) [90]. In practice, we allowed the following set of transformations: pixel blitting (x-flip, 90-degree rotation, integer translation), general geometric transformation (isotropic scaling, anisotropic scaling, fractional translation), and color transformation (brightness, luma flip, hue, saturation). The total length of training of StyleGAN2-ADA is defined by "the total number of real images", since the randomization of transformation is done separately for each image in a minibatch. We trained the model on one Tesla V100 GPU for a total length of 3,836,000 images, using the recommended learning rate of 0.002 and $R_1$ regularization of 10 [90] for generating 1024 pixels × 1024 pixels resolution outputs. The FID (Fréchet Inception Distance), KID (Kernel Inception Distance), and recall for the trained model are 13.07, 0.0038, and 0.330 respectively.

**pixel2style2pixel (pSp) encoder.** Upon training the StyleGAN2-ADA, we separately trained a pSp encoder on 80% of randomly sampled images from the TID dataset and validated on the rest of the images. We implemented the pSp encoder based on the code released by https://github.com/eladrich/pixel2style2pixel [91]. The pSp encoder aims to efficiently embed a real photo into StyleGAN's extended intermediate latent space $W+$ [96]. Unlike $W$ space, $W+$ is a concatenation of 18 different 512-dimensional vectors ($w_1$ to $w_{18}$), one for each convolution layer of StyleGAN2-ADA generator. Given a real image, we can map it to the latent space $W+$ and create its reconstruction image by feeding the obtained latent code into our pretrained StyleGAN2-ADA generator.

The pSp encoder is built on a feature pyramid network [122] to generate three levels of feature maps (coarse, medium, and fine) [88] from which 18 latent vectors of $W+$ were extracted using a small fully convolutional network (map2style). Latent vectors $w_1$ to $w_3$ are generated from the small feature map, $w_4$ to $w_7$ are generated from the medium feature map, and $w_8$ to $w_{18}$ are generated from the large feature map. The latent vectors are then injected into the pretrained StyleGAN2-ADA generator corresponding to their spatial scales to synthesize the reconstructed image. The feature pyramid network and the map2style networks are updated through backpropagation to learn to generate latent vectors which map to reconstructed images that are perceptually similar to the input real images. The architecture is illustrated in Fig 1B.

The entire framework was trained on a set of loss functions to encourage the accurate reconstruction of the real photos: pixel-wise loss ($L_2$), LPIPS loss ($L_{LPIPS}$), and regularization loss ($L_{reg}$). For an input image $x$, the total loss is defined as: $L(x) = \lambda_1 L_2(x) + \lambda_2 L_{LPIPS}(x) + \lambda_3 L_{reg}(x)$, where $\lambda_1$, $\lambda_2$, and $\lambda_3$ are constants defining the loss weights. Here, we set $\lambda_1 = 1$,

$\lambda_2 = 0.8$, $\lambda_3 = 0.005$. The maximum number of training steps was set at 10000, and the model leading to the minimum total loss was consistently updated. We trained the model with one Tesla V100 GPU for 2 GPU days, and the model optimized at training step 9000 was used for the rest of the study. The total loss was 0.181.

## Image generation with DCGAN

We also explored the feasibility of learning to synthesize images of translucent objects with a non-style-based generative adversarial model, DCGAN, whose generator only takes the input from a uniform noise distribution $Z$ (i.e., input latent space), and gradually applies a series of fractionally-strided convolutions to obtain up-sampled feature maps. DCGAN has demonstrated the capability of generating reasonable results from a variety of datasets with 32 to 64 resolution images, such as indoor bedrooms and human faces [93]. We used the DCGAN architecture proposed by Radford et al. (2015) to train to generate 64 pixels × 64 pixels images of soaps. The DCGAN is trained on the TID dataset, with images resized to 64 pixels × 64 pixels. The model was trained for 800 epochs, with the following hyperparameters: the input latent space $Z$ has dimension $(100 \times 1)$, the learning rate is 0.0002, the batch size is 128, and the momentum is 0.5 for the Adam optimizer. Although DCGAN captured some degree of variation of translucency, it fails to accurately depict the shapes of the objects. In general, the generated results have much poorer perceptual quality in comparison to those from StyleGAN (see Supplementary S8 Fig).

## Psychophysical experiments

**Participants.** The same group of twenty participants completed Experiments 1 and 2 (N = 20, median age, 20; age range, 18–27, 12 female, 8 male). They completed the experiments in one lab-based session. Another group of twenty participants completed Experiment 3 (N = 20; median age, 21; age range, 18–27; 10 female, 10 male). Five individuals participated in all experiments. Observers received no information about the hypotheses of the experiments. No statistical methods were used to predetermine sample sizes, but our sample sizes are consistent with those reported in previous publications of material perception measured in the laboratory [69, 80, 87]. All observers had normal or corrected-to-normal visual acuity and normal color vision. Participants were primarily undergraduate students from American University. The observers were given written informed consent and were compensated with either research course credits from American University or with $16 per hour.

**Psychophysical procedures.** The psychophysical experiments were conducted in a dimly lit laboratory room. Observers sat approximately 7 inches away from the monitor and were given no fixation instructions. The stimuli were presented on an Apple iMac computer with a 27-inch Retina Display with a resolution of 5120 pixels × 2880 pixels and a refresh rate of 60 Hz. PsychoPy v.2021.1.2 was used to present the stimuli and collect the data [123]. At the beginning of each experiment, observers were given experiment-specific instructions and demos.

**Experiment 1: Real-vs-generated discrimination.**

**Stimuli.** To avoid using the same images as those in the model training process, we took 300 new photographs of our collection of soaps. We then split these photographs equally into two groups (A and B), which similarly capture the variety of materials, lighting fields, and viewpoints. The 150 real photographs from Group A and the 150 generated images obtained from Group B were used as stimuli for Experiment 1. Specifically, photographs from Group B were first encoded into the $W+$ latent space through the pSp encoder, and then were reconstructed through our trained StyleGAN2-ADA generator. In this way, we obtained generated

images that cover the diverse samples of appearances of soaps in our dataset. Examples of stimuli are shown in Fig 2A. All images were presented in size 1024 pixels × 1024 pixels.

**Experimental procedure.**    We first gave a brief introduction to each observer of how the real photographs and generated images of soaps were created. The observers were told that the "Real photographs of the soaps (Real) were taken using a smartphone camera, and the generated images were produced from a computer algorithm (Generated). The generated images would try to resemble the visual appearances of the object in the real photos." Afterward, the observers were presented with a series of images and were asked to judge whether the stimulus is Real or Generated. Each image was shortly displayed for one second, and then the observer made the judgment with a key press. Observers were given the prior knowledge that 50% of the stimuli were Real. We conducted the experiment with two repeats. In repeat 1, observers judged 300 images of real and generated images with a pre-randomized order in two blocks of 150 trials. They then completed another repeat of the same 300 images but with a different pre-randomized order. The experimental procedure is shown in Fig 2B.

**Experiment 2: Material attribute rating.**

**Stimuli.**    Experiment 2 stimuli were the same 300 images of real photographs and model-generated images of soaps as in Experiment 1.

**Experimental procedure.**    Before the experiment started, we introduced the concept of translucency to the observers by showing them a simplified illustration of the subsurface scattering process (Supplementary S2 Fig). In Experiment 2, observers were asked to rate the material attributes of the images. On each trial, the observers rated each attribute using a seven-point scale (7 means high, 1 means low) by adjusting the slider (Fig 3A). They had unlimited time to make judgments. The 300 images were equally split into two blocks, and presented in a pre-randomized order. This experiment was conducted with only one repeat.

Observers were provided with the definition of the material attributes as the following:

- Translucency: To what degree the object appears to be translucent.

- See-throughness: To what degree the object allows light to penetrate through.

- Glow: To what degree the object appears to glow light from inside.

While "see-throughness" is a visual term that might have a more common understanding among observers, judging "translucency" might depend more on individual interpretation and provide additional/complementary insights into the material qualities. Thus, both attributes were measured in this experiment.

**Experiment 3: Perceptual evaluation of emerged scene attributes.**

**Stimuli.**    We created image sequences by applying morphing between the source image A ($w_A$) and the target image B ($w_B$) using Eq 1. The morphing was separately applied on three sets of layers of the latent space: early-layers (layers 1 to 6), middle-layers (layers 7 to 9), and later-layers (layers 10 to 18), with equal interpolation steps. To generate an image sequence, the interpolation step (λ), was set to have four values: 0, 0.33, 0.67, and 1 (see Fig 4A).

We picked 24 soaps from the TID dataset; half were opaque milky soaps (generally with low translucency, i.e., opaque) and half were translucent glycerin soaps (generally with high translucency, i.e., translucent). With these images of soaps, source-target image pairs were formed under three conditions: opaque-translucent (OT), opaque-opaque (OO), and translucent-translucent (TT). For each condition of source-target pairs, we created image sequences based on the morphing on the early-, middle-, and later-layers respectively, and then randomly sampled 50 sequences as stimuli. This led to 3 (condition of source-target pair) × 3 (layer-manipulation method) × 50 image sequences in total (Fig 4A). All individual images in the image sequences were resized to 256 pixels × 256 pixels for display.

**Experimental procedure.** At the beginning of the experiment, we showed observers a few samples of real soaps of different materials, shapes, and body colors with the goal of illustrating the effects of these scene attributes on material appearance.

The observers viewed 450 image sequences. For each image sequence, observers selected the "One most prominent visual attribute changed from left to right" from one of the following: shape/orientation, color, material (e.g., translucency), and lighting. The image sequences were equally split into three blocks and presented in a pre-randomized order. Observers had unlimited time to complete their judgment on each trial (Fig 4B). This experiment was conducted with one repeat.

## Computing translucency decision boundaries from latent code

We trained binary SVM based on the latent vector of each layer of the latent space $W+$ to classify the material of the soap in the TAG-generated image as either "milky" or "glycerin". The trained SVM classifiers were then used to generate model predictions on a continuous scale. We randomly sampled 500 real photographs of "milky" soaps and another 500 photos of "glycerin" soaps from the TID to train the SVM. The 1000 photos were first embedded into the $W+$ latent space to obtain their corresponding $18 \times 512$ dimensional latent codes through our trained pSp encoder. Since the latent space contains 18 layers, we trained 18 SVM models based on each layer's latent vectors of the embedded images. In other words, there were 18 different feature matrices, each with the dimension of $n \times 512$, where $n$ is the number of training samples. We implemented SVM from *scikit-learn* with nested cross-validation for model fitting, and used a relatively strong regularization ($C \in [0.001, 0.1]$) to reduce overfitting [124]. Hence, we obtained a linear decision boundary $d_i$ for the i-th layer's latent vectors. We then computed the model prediction values of the 150 generated images used in Experiment 2. With the $w_+ \in W+$ latent code of a generated image, we extracted its i-th layer's latent vector and measured its distance from $d_i$. For each layer, the model prediction value, which is the normalized distance from $d_i$, was compared to human perceptual rating data from Experiment 2.

## Pixel-based image embeddings from dimensionality reduction methods: t-SNE and MDS

We computed embeddings of the raw images of translucent materials by applying dimensionality reduction methods. T-SNE, a non-linear algorithm, seeks to find a faithful low-dimensional embedding of the data points from the high-dimensional space, while preserving the structure of data as much as possible [125]. MDS aims to project the high-dimensional data into a lower-dimensional space that preserves the pairwise distances between data points.

We implemented t-SNE and MDS using *scikit-learn* package in Python. For both methods, we set the dimension of the embedded space at 512 and the maximum number of iterations at 300. For t-SNE, we experimented with perplexity at 5, 15, and 25. For MDS, we used Euclidean distance as the measurement of dissimilarity between data points. The results of the model predictions of human translucency attribute ratings are shown in S9 Fig in Supplementary material.

## Independent Component Analysis (ICA) for the intermediate generative representation

Based on the results of Experiment 2, we selected 40 generated images with the highest translucency ratings (high-translucency). Meanwhile, we selected another 40 generated images of soaps with various shapes, orientations, and lighting environments. Then, we fully paired

these 80 images (source) with the 40 high-translucency images (target). To create a new "high" translucency image, we replaced the middle-layer (layers 7 to 9) latent vectors of the source image with those of the target and used the resultant latent code to generate the corresponding image through the generator. Then, we extracted the intermediate generated result from the tRGB layer corresponding to 64 pixels × 64 pixels spatial scale of which translucency is established. We repeated this step to obtain 3160 "high" translucency images at a resolution of 64 pixels × 64 pixels (Fig 7A). For each image in the "high" translucency dataset, we first resized it to 512 pixels × 512 pixels resolution and sampled 10 image patches of 24 pixels × 24 pixels from random locations. FastICA was then applied on the 3160 × 10 image patches to learn 64 basis functions (i.e., middle-layer ICA kernels) [106]. For the learning of middle-layer ICA kernels, we also conducted the FastICA with different samplings of the image patches with 64 and 100 components (Supplementary S6 Fig).

## Statistical analysis

We used Bayesian multilevel multinomial logistic regression to model the psychophysical results from Experiment 3 [100, 126]. The goal is to examine whether the prominent scene attributes judged by the observers can be predicted by the layer-manipulation methods. We implemented the *brms* library supported in *R* for the analysis. The model's dependent variable is the scene attribute (i.e., shape/orientation, color, material, and lighting). The predictors include the layer-manipulation methods (i.e., early-layer manipulation, middle-layer manipulation, and later-layer manipulation), the type of source-target pair (i.e., opaque-opaque (OO), opaque-translucent (OT), and translucent-translucent (TT)), and the interaction between these two factors, while considering the individual observer as a grouping variable. Three Markov chains were used for the parameter posterior distribution estimation, with 8000 iterations for each chain of the Markov Chain Monte Carlo (MCMC) algorithm. We assumed a uniform distribution for the priors of the parameters. The complete results of the analysis can be found in Supplementary S3 Fig, S1 Table, and S1 Text.

## Supporting information

**S1 Text. Explanatory texts for each figure and table in Supporting Information.** (PDF)

**S1 Fig. Example stimuli from the real-versus-generated discrimination experiment agreed by the majority of observers.** Each image is resized for display. (TIF)

**S2 Fig. Illustrations of the simplified light transport process.** Left: light transport process for an opaque object. Right: subsurface scattering for a translucent object. (TIF)

**S3 Fig. Conditional effect of layer-manipulation method on the prediction of scene attribute using Bayesian multilevel multinomial logistic regression model.** The x-axis is the layer-manipulation method, and the y-axis is the estimated probability that a certain scene attribute is selected as the most prominent attribute that has been changed in an image sequence. The error bar indicates the upper and lower bounds of the estimation at the confidence level of 95%. The panels show the predicted results for three source-target pair conditions: opaque-translucent (OT), opaque-opaque (OO), and translucent-translucent (TT). (TIF)

**S4 Fig. Manipulating the i-th layer's latent vector of the original image (leftmost) along the positive direction of the normal of the learned translucent decision boundary ($d_i$).** The displacement on the middle-layers (layers 7 to 9) can mainly affect the translucent appearance. (TIF)

**S5 Fig. Manipulating the i-th layer's latent vector of the original image (leftmost) along the negative direction of the normal of the learned translucent decision boundary ($d_i$).** The displacement on the middle-layers (layers 7 to 9) can mainly affect the translucent appearance. (TIF)

**S6 Fig. Middle-layer ICA kernels extracted from intermediate activation of high-translucency images (see main paper Fig 7).** Top and bottom rows show the FastICA results of using 64 and 100 components respectively. Within each row, each panel shows the kernels learned from a different random sampling of the image patches. The kernels are $24 \times 24$ and are resized for display. (TIF)

**S7 Fig. Visualization of features captured in the early-layers of the learned latent space.** (A) The intermediate generated results (tRGB layer output at 16 pixels × 16 pixels resolution) of the images from the high-translucency dataset. The images are resized for display. (B) Early-layer ICA kernels obtained by training a system of 64 basis functions on 96 pixels × 96 pixels image patches extracted from images in (A). The kernels are of size 96 × 96. (C) Visualization of applying three-dimensional convolution of the individual early-layer ICA kernels in (B) on a real photograph of translucent soap. (D) The resulting filtered images of four different soaps with selected chromatic and achromatic kernels. (TIF)

**S8 Fig. Examples of DCGAN-generated results.** (A) Examples from the training dataset, which are images from the TID dataset, resized to 64 pixels × 64 pixels. (B) Examples of DCGAN-generated soaps after 800 epochs of training. (TIF)

**S9 Fig. Image embeddings obtained from applying dimensionality reduction methods on raw images fail to predict human ratings of translucency-related attributes.** The scatter plots show the model prediction value computed from the embedding of images using dimensionality reduction methods, with the correlation coefficients (correlation between the model prediction and human perceptual ratings, $r_{hc}$) and the corresponding $p$-values. Green, blue, and orange colors represent the data of translucency, see-throughness, and glow, respectively. (TIF)

**S1 Table. Summary of Bayesian multilevel multinomial logistic regression model outputs.** The leftmost column shows the name of the parameter. The names of the response variable and the predictor are separated by "_". The second to fifth columns are the exponentiated mean (Mean Est), the standard error (Est.Error), and the lower (HDI Lower) and upper bounds (HDI Upper) of the 95% credible interval of the posterior distribution for each parameter. The last column is the percentage of the 95% HDI of parameter distribution that falls inside the region of practical equivalence (ROPE). (TIF)

**S2 Table. Correlation coefficients ($r_{hc}$) between each layer's model prediction and the perceptual ratings.** The table shows the Pearson correlation between the model predictions and

the mean normalized attribute ratings from Experiment 2 (columns 2 to 4), with their corresponding *p*-values.
(TIF)

## Acknowledgments

We thank Eric Schuler for the valuable discussion on the statistical analysis of this work and Alex Godwin for the discussion of data visualization.

## Author Contributions

**Conceptualization:** Chenxi Liao, Masataka Sawayama, Bei Xiao.

**Data curation:** Chenxi Liao.

**Formal analysis:** Chenxi Liao, Bei Xiao.

**Funding acquisition:** Bei Xiao.

**Investigation:** Chenxi Liao, Masataka Sawayama, Bei Xiao.

**Methodology:** Chenxi Liao, Masataka Sawayama.

**Project administration:** Chenxi Liao, Bei Xiao.

**Resources:** Chenxi Liao, Bei Xiao.

**Software:** Chenxi Liao.

**Supervision:** Bei Xiao.

**Validation:** Chenxi Liao, Masataka Sawayama.

**Visualization:** Chenxi Liao, Masataka Sawayama.

**Writing – original draft:** Chenxi Liao, Bei Xiao.

**Writing – review & editing:** Chenxi Liao, Masataka Sawayama, Bei Xiao.

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
