## [Decision Letter · Decision Letter 0]

20 Dec 2022

Dear Mr Liao,

Thank you very much for submitting your manuscript "Unsupervised learning reveals interpretable latent representations for translucency perception" for consideration at PLOS Computational Biology. As with all papers reviewed by the journal, your manuscript was reviewed by members of the editorial board and by several independent reviewers. The reviewers appreciated the attention to an important topic. Based on the reviews, we are likely to accept this manuscript for publication, providing that you modify the manuscript according to the review recommendations.

Sincerely,

Roland W. Fleming, PhD

Academic Editor

PLOS Computational Biology

Thomas Serre

Section Editor

PLOS Computational Biology

Reviewer's Responses to Questions

**Comments to the Authors:**

Reviewer #1: This paper reports unsupervised learning of translucency. Overall, I find the paper quite thought provoking and the writing to have above average clarity.

The stimuli are rectangular prisms or ellipsoids made from various forms of soap that strongly vary in translucency. My impression is that nearly all of the example surfaces shown in the paper’s figures appear to have strong back lighting. I think it is worth commenting in the main text about the distribution of illumination directions in the set of images used to train the network.

The experiments that are particularly interesting manipulate the middle layers of the network, which appears to encode information strongly related to material appearance. Specifically, the results suggest that middle layer 9 encodes information about opacity and translucency. I find that result very interesting, and it is at this point in the paper that I desired the text to provide more detail:

The basic idea is to interrogate what the network has learnt by editing the information in one layer and observing the effect on the synthesised image. Say, the network’s input image is a bar of opaque soap. All of the information needed to reproduce that image is held fixed except the information in the layer whose “function” is being interrogated. The information in that layer is then replaced with the same layer’s state when the input is a different surface, say a bar of translucent soap. The interesting result is that the network now synthesises an image of a surface that has the same 3D shape as the opaque soap but appears translucent. It is at this point in the text that I think it would be useful to comment on the extent of the differences between the two surfaces that are being “mixed together”. How different are the overall sizes of the surfaces, their 3D shapes, colors, and illumination contexts?

The only other question I have is whether all the materials and code needed to reproduce this interesting result will be made available and easily accessible?

Reviewer #2: SUMMARY

A beautifully illustrated paper presenting a very successful technical achievement in training a generative DNN to learn the visual appearances of simple objects varying in shape, colour, and translucency. The network is trained using a custom dataset of real photographs (rather than the easier-to-collect rendered images more often used in similar work) and can generate images at an exceptionally high resolution (1024x1024 pixels). Three human behavioural experiments confirm that generated images are in many cases indistinguishable from real photographs (Expt 1), span a similar range of apparent material properties (Expt 2), and can be made to morph coherently along different appearance dimensions (Expt 3).

MAIN COMMENTS

I'm not sure how surprising is it that "human-understandable scene attributes emerge" in the latent code...this has been shown several times in several domains for GAN-based DNNs (e.g. with faces, chairs, landscapes), and is arguably necessarily true of any model that can smoothly interpolate a series of plausible objects morphing between two existing objects. Likewise with the analysis of spatial scales: it seems intuitive that small-scale layers contain information about colour (which is present within the RGB values of even small kernels), whereas coarse-scale layers specify shape (a whole-object attribute); does this tell us anything new?

This makes me wonder whether the authors satisfy their stated goal of "discovering perceptually relevant image features" underlying translucency perception. The network does indeed seem to have learned these features in a summary code that the authors are able to navigate; but do we end up with a clearer description of *what* these features are than we had before? Specifically, it would be wonderful to provide evidence that the representation of translucency is "human-like", rather than (just) optically accurate. e.g. perhaps by testing for quirks of human translucency perception such as the lower apparent translucency of greyscale images.

Having said that, I think the paper works as an inspiring demonstration of the potential value of carefully-curated generative networks in perception science and psychological research more broadly. The combination of network architectures (StyleGAN + a pixel-style-pixel encoder to project new images into its latent space) is well-chosen - they have been little-used in the perception science literature but solve the challenge of training on this limited dataset excellently and provide a navigable and interpretable latent space. Despite the very constrained application domain (a dataset of photographs of 60 individual soap objects), this in itself is a useful technical contribution to the field.

The authors make reasonable and restrained interpretations of the results. They explicitly avoid claiming that either the architecture or learning process of the StyleGAN-based TAG network constitute good mechanistic models of human material perception. Instead they focus on the network's value as tool for data-driven discovery of complex image features that are candidates for forming the perceptual dimensions of material perception. I find this an admirably cautious and nuanced use of DNNs, and I'm sympathetic to their conclusion that "learning the scale-specific statistical structure of natural images might be crucial for...material properties".

MINOR COMMENTS

1. I'm a bit skeptical of the claim made in the Discussion (around line 307) that models without a multiscale representation (e.g. DCGAN) aren't capable of capturing translucency. Previous work shows that GANs trained on rendered images can reasonably well caputure glass-like transparency: Tamura, Prokott & Fleming (2022) https://jov.arvojournals.org/article.aspx?articleid=2778652. And "Stable Diffusion" type models currently show an outstanding ability to render the nuances of materials, including translucency - do these also have explicit multi-scale latent codes?

2. Why ask both about "translucency" and about "see-throughness"; are these distinct concepts? The explanation given (that these were "found to be descriptive" of translucent objects in a previous study by Liao, Sawayama & Xiao (2022 JoV) doesn't fully explain it, since that study doesn't seem to have asked people to rate the two dimensions simultaneously, but rather asked for a binary translucency judgement followed by a continuous "see-throughness" judgement, and found these were closely related.

TYPOS etc

The paper generally reads very fluidly. There were just a few phrases that read oddly to me, e.g. from the first couple of pages:

- line 35: "difference between raw and readily cooked food"

- line 43: "...and in the mean time, humans may lack precise descriptions"

- line 61: "Some recent works in perceptual system..."

Reviewer #3: This paper presents an unsupervised approach that reveals a layered representation of perceived translucency, at least for the subset of cases included in the study. It was a pleasure to read, the authors motivate their decisions convincingly enough and summarize their main findings as they move along, which always helps the reader.

I found the methodology solid, the write-up excellent, the figures well explained and informative. The perception of materials in general, and translucency in particular, is an open topic spanning different fields of research, from neuroscience to computer graphics and even robotics. I believe the paper makes a useful contribution in this regard. I found the resulting layered space intriguing, and it seems to work well.

I don't usually review nor read papers in this journal, so it may be that I'm miscalibrated, but all in all I think this is a very good paper.

I only have a few questions, comments and suggestions.

Fig. 9 in the supplemental is used by the authors to illustrate how other architectures (DCGAN) fail to generate good results. However, other than the shape, the results (for the most part) seem to actually be pretty decent regarding translucency. It'd be great if the authors could offer more insights about this.

The user study at the beginning (summarized in Fig. 2): Isn't one second too short to view the images? I wonder if the results would've varied giving the users a bit more time to analyzed the images in a more relaxed way. If possible, extending the test to maybe 5 seconds would be useful (maybe with a randomized subset of images not to make it too long)

As with most papers, the results and conclusions are strictly speaking only valid for the particular subset of examples used. In that sense, and this is really my only "major" concern, some parts of the paper including the title may be a bit overstated. The authors have used only one class of translucent materials only, with basic geometries and one particular, strongly angled lighting (to emphasize the translucency effect). How do the results generalize to other setups? This would probably need to be discussed at the end of the paper, possibly showing some failure examples along with some insights as to why they may occur.

A couple of relevant papers on the topic of discovering perceptual spaces for materials could be added:

Pellacini et al. Toward a psychophysically-based light relfection model for image synthesis. SIGGRAPH 2000

Serrano et al. An intuitive control space for material appearance. ACM Transactions on Graphics, SIGGRAPH Asia 2016

**Have the authors made all data and (if applicable) computational code underlying the findings in their manuscript fully available?**

Reviewer #1: None

Reviewer #2: Yes

Reviewer #3: **No: **Maybe they did. Sorry, but I had to review the paper on a train and couldn't check.

PLOS authors have the option to publish the peer review history of their article (what does this mean?). If published, this will include your full peer review and any attached files.

Reviewer #1: No

Reviewer #2: **Yes: **Katherine Storrs

Reviewer #3: No

Figure Files:

Data Requirements:

Reproducibility:

References:

---

## [Editor Report · Decision Letter 1]

18 Jan 2023

Dear Mr Liao,

We are pleased to inform you that your manuscript 'Unsupervised learning reveals interpretable latent representations for translucency perception' has been provisionally accepted for publication in PLOS Computational Biology.

Best regards,

Roland W. Fleming, PhD

Academic Editor

PLOS Computational Biology

Thomas Serre

Section Editor

PLOS Computational Biology

---

## [Editor Report · Acceptance letter]

3 Feb 2023

PCOMPBIOL-D-22-01676R1 

Unsupervised learning reveals interpretable latent representations for translucency perception

Dear Dr Liao,

I am pleased to inform you that your manuscript has been formally accepted for publication in PLOS Computational Biology. Your manuscript is now with our production department and you will be notified of the publication date in due course.

With kind regards,

Anita Estes
